# Uptake of Fertilizer Nitrogen and Soil Nitrogen by Sorghum Sudangrass (*Sorghum bicolor* × *Sorghum sudanense)* in a Greenhouse Experiment with $^{15}$N-Labelled Ammonium Nitrate

Lucas Knebl [1,2,*], Andreas Gattinger [1], Wiebke Niether [1] and Christopher Brock [1,2]

1   Department of Agronomy and Plant Breeding II, Justus Liebig University Giessen, Karl-Glöckner Str. 21C, 35392 Giessen, Germany; andreas.gattinger@agrar.uni-giessen.de (A.G.); wiebke.niether@agrar.uni-giessen.de (W.N.); brock@forschungsring.de (C.B.)
2   Forschungsring e.V., Brandschneise 5, 64295 Darmstadt, Germany
*   Correspondence: knebl@forschungsring.de

**Abstract:** A greenhouse experiment with sorghum sudangrass (*Sorghum bicolor* × *Sorghum sudanense*) and maize (*Zea mays*) was conducted to assess information on differences in their nitrogen and fertilizer utilization when used as energy crops. The aim was to contribute to the scarce data on sorghum sudangrass as an energy crop with regards to nitrogen derived from fertilizer (NdfF) in the plant's biomass and fertilizer nitrogen utilization (FNU). Sorghum sudangrass and maize were each grown in eight bags of 45 L volume and harvested at maturity after 154 days. Each crop treatment was further divided in a control treatment (four bags each) that did not receive N fertilization and a fertilization treatment (four bags each) that received 1.76 g N, applying a $^{15}$N-labelled liquid ammonium nitrate fertilizer. Fertilization took place at the start of the experiment. After harvest, the whole plant was divided in the fractions "aboveground biomass" (ABM) and "stubble + rootstock" (S + R). Weight, N content and $^{15}$N content were recorded for each fraction. In addition, N content and $^{15}$N content were assessed in the soil before sowing and after harvest. The experiment showed that FNU of sorghum sudangrass (65%) was significantly higher than that of maize (49%). Both crops accumulated more soil N than fertilizer N. The share of fertilizer N on total N uptake was also higher with sorghum sudangrass (NdfF = 38%) compared to maize (NdfF = 34%). The observations made with our control plant (maize), showed that the results are plausible and comparable to other $^{15}$N studies on maize regarding yields, NdfF, and FNU, leading to the assumption that results on sorghum sudangrass are plausible as well. We therefore conclude that the results of our study can be used for the preliminary parametrization of sorghum sudangrass in soil organic matter (SOM) balance at field level.

**Keywords:** mineral nitrogen utilization; $^{15}$N-labelled fertilizer; N balance; sorghum sudangrass

## 1. Introduction

The appropriate management of soil organic matter (SOM) is a major goal for sustainable agriculture [1]. Farmers must be able to estimate SOM contribution by demand of specific crops in order to adapt crop rotation and nutrient cycling to ensure optimal SOM reproduction [2]. Nitrogen is an important compound of SOM and a necessary nutrient for plants. However, even with standard mineral N fertilization that is adjusted to meet their demand, the total N uptake by crops consists of soil-derived N (NdfS) next to fertilizer-derived N (NdfF). Those soil-derived N amounts have to be replaced in order to maintain soil fertility. Fertilizer N that has not been taken up can either remain in the soil or be lost (e.g., leaching, volatilization). Fertilizer nitrogen utilization (FNU) provides an important indication of the potential amount of fertilizer N that remains in the soil (and may be available to the following crop) or is lost.

One way to estimate FNU of a crop is to use a $^{15}$N-labelled fertilizer and thus record NdfF within the plant. Such experiments exist mainly for maize and cereal [3–13]. The studies on maize differ considerably in design and implementation, making comparability difficult. Differences exist in the fertilization (type, amount, fertilization timing, and application technique). In addition, the focus of the assessments often is on aboveground biomass (either in total or varying plant fractions). Fertilizer uptake by roots and remaining fertilizer in the soil is often neglected. Among available studies, only a few are comparable to our experimental setup based on fertilizer type (NH$_4$ and/or NO$_3$), crops, and sample size. A list of those studies can be found in Appendix A Table A1. Harris et al. [4] conducted a study on maize, examining grain, stem, and roots as well as soil after one growing season. In their experiment, 124 kg N*ha$^{-1}$ was fertilized prior to seeding (granular, incorporated) in the form of $^{15}$N-labelled ($^{15}$NH$_4$)$_2$SO$_4$. The authors found that corn had an overall FNU of 40%, while 23% of the applied fertilizer remained in the soil after harvest and about 38% of applied fertilizer could not be recovered in the samples taken and was assumed to possibly be lost. No data were provided on NdfF. Studies on maize that considered residual fertilizer N in the soil but not in the roots show comparable results to other studies on maize [3,4,6–8]. The NdfF reported in these studies ranged from 11% to 74% (median = 31%) and FNU of aboveground biomass ranged from 24% to 62% (median = 43%). Fertilizer N remaining in the soil in these studies was 14% to 46% of the original amount of N applied (median = 25%). Porter [6] only checked the amount of labelled N in the soil solution. Measured residual labelled NO$_3$ ranged from 1% to 10% of the applied fertilizer amount. The proportion of fertilizer N that was not recovered (see Appendix A Table A1) shows a high variance in the studies, ranging from 15% to 76% (median = 29%). In this regard, a positive relationship was observed between fertilizer quantity, and fertilizer N that could not be recovered [6].

However, some crops only recently received attention (<20 years), for example, plants cultivated for energy use. Against the background of climate change, plants for energy production are increasingly coming into focus. Plants that still achieve good yields under extreme conditions (e.g., drought, heat) and need low N input are advantageous in this regard. Sorghum species are characterized by comparatively good water use efficiency, tolerance to higher temperatures, lower fertilizer requirements, and advantages in erosion and weed control [14–16]. Under limited water availability and high temperature, sorghum species performance is higher than that of maize and other cereals [17]. Available studies in this regard focus on sorghum species for forage production, especially in arid and semiarid regions [16–21]. The species *Sorghum bicolor* × *Sorghum sudanense* (from here on called sorghum sudangrass) is particularly suitable and therefore commonly cultivated for bioenergy use. When growing this hybrids, farmers should be able to estimate N use efficiency in order to maximize yields without depleting their agricultural soils. However, for this crop, no long-term field experiments exist yet that can deliver a well-founded data base.

In cases without long-term observations, SOM balance models such as Roth-C, CCB, and HU-MOD [22,23] can provide first clues on how a specific crop might affect SOM. Usually, SOM balance models provide a decision tool in planning management strategies with regard to SOM maintenance or enhancement. Brock et al. [23] reviewed most commonly used SOM balance models (in the EU) in detail. Depending on the model, the specific effects of a cultivation system on the balance of soil organic carbon (SOC) and/or soil total nitrogen (STN) are calculated. The algorithms used are usually based on data derived from long-term field experiments on the corresponding cropping systems. The respective effects of individual crops are determined by the corresponding C and/or N inputs and withdrawals. If the data on one crop are scarce, SOM balance models can use data from an already studied physiologically similar plant. In the case of sorghum sudangrass, this could be *Zea mays* to make an initial estimate. However, the approach could as well result in a significant overestimation or underestimation of the true effects of sorghum sudangrass on SOM. Therefore, the aim of this study is to contribute to expanding the data situation

on sorghum sudangrass as a bioenergy crop with regard to its fertilizer utilization (NdfF and FNU). In addition, we want to know whether a greenhouse experiment is sufficient to assess data that can be used to model SOM balance of crops in the field. The hypothesis of this study is that there are significant differences between sorghum sudangrass and maize regarding NdfF and FNU. This assumption is based on the reported differences between sorghum species and maize with regard to fertilizer use [14–17]. We further hypothesized that those differences can be assessed in a greenhouse experiment. For this purpose, sorghum sudangrass and maize were grown in a bag experiment in a greenhouse and fertilized with ammonium nitrate, labelled with $^{15}$N to assess NdfF and FNU.

## 2. Materials and Methods

### 2.1. Experimental Set Up

The experiment was carried out in 2015 at the Experimental Station Rauischholzhausen of the Justus Liebig University Giessen, Germany (Mean annual temperature: 9 °C; geographic location: 50°75′79.7″ N 8°88′75.1″ E). Sorghum sudangrass (*S. bicolor* × *S. sudanense*) and maize (*Zea mays*) as control plant were grown in eight planting bags each (four fertilized, four not fertilized) (Table 1). Planting bags had a volume of 45 L, with a surface area of 0.126 m$^2$ and an approximate height of 35–36 cm. The trial started in mid-March with sowing and ended in mid-August with harvest. The planting bags were placed randomly in a greenhouse with daylight, which could be opened to the outside. Several holes at the bottom of the bags allowed for possible leaching. A standard soil substrate of the experimental station was used to fill the bags. The substrate consisted of an arable topsoil (stored air dry for a longer period prior to the experiment) that was sieved to <5 mm and mixed with quartz sand (2:1) when the experiment started. The mixture had a SOC and STN content of 0.95% and 0.10%, respectively, with a pH of 6.3. Plant available nitrogen ($N_{min}$) in the substrate corresponded to 200 kg $N_{min}$*ha$^{-1}$. Plant bags were filled with soil up to 36 cm height (compacted to 33 cm at harvest). The upper 15 cm were mixed with fertilizer homogeneously before filling the bag. Fertilized treatments (N1) received 1.76 g of nitrogen with liquid $^{15}$N-labeled ammonium nitrate (50% ammonia, 50% nitrate, 1% enrichment). The fertilizer N amount corresponds to a fertilization of 140 kg N ha$^{-1}$. Control treatments (N0) did not receive any fertilization. Sorghum sudangrass (variety "Lussi") and maize (variety "Lorado") were sown immediately after filling the bags at a seed depth of 3 cm and 4 cm, respectively. For this purpose, five plants per bag were sown in four replicates each. For sorghum sudangrass, this seeding density and the N fertilization were in accordance with standard practice. Maize was seeded more densely than in practice, and the N fertilization was relatively low given the expected increased yields (due to the dense seeding). Seeding density for maize was five times higher than usually used in practice. This was applied to obtain a more homogeneous plant material (instead of only one plant). Seeds that did not germinate initially were replanted as soon as possible to reach five plants per bag at harvest time. Maize is a crop that has been sufficiently studied scientifically and the effects of seed density and N fertilization can be well estimated and evaluated. Plants were irrigated according to plant demand. The harvest of both crops took place after 154 days when plants had a dry matter content of 28–32%.

**Table 1.** Experimental setup of $^{15}$N experiment with sorghum sudangrass and maize grown in bags (45 L vol.) in a greenhouse. DM = dry matter.

| Plant | Treatment (ID) | Repetition (No.) | Fertilization (g N*Bag$^{-1}$) | Seed Density (Seeds*Bag$^{-1}$) | DM at Harvest (%) | Harvest (Day) |
|---|---|---|---|---|---|---|
| Sorghum sudangrass (*S. bicolor* × *S. sudanense*) | N0 | 4 | 0 | 5 | 30 ± 2 | 154 |
| | N1 | 4 | 1.76 | 5 | 30 ± 2 | 154 |
| Maize (*Zea mays*) | N0 | 4 | 0 | 5 | 30 ± 2 | 154 |
| | N1 | 4 | 1.76 | 5 | 30 ± 2 | 154 |

### 2.2. Data Collection and Analysis

After sieving the soil substrate and mixing it with sand, three subsamples were taken before the substrate was fertilized and divided to be filled into the planting bags. After harvest, 3 soil samples per bag were taken with a soil auger (0–30 cm). The samples were oven-dried directly after collection at 40 °C for a period of 48 h, sieved to <2 mm, and grinded for analysis of STN, and $^{15}N$ content. At harvest, aboveground biomass (ABM) of the plants was cut at 10 cm height. To assess the main part of residues that stay on the field after harvest, stubble and rootstock of all plants within each bag were combined to the fraction stubble + rootstock (S + R). For total biomass (TBM), both plant fractions were summed up. Soil from rootstocks was removed by washing the rootstock with tap water over a sieve. Plant fractions were immediately oven-dried at 60 °C for a period of 48 h and grinded to be analyzed for carbon (C), nitrogen (N), and $^{15}N$ content. N content in both soil and plant biomass fractions, as well as organic carbon in soil, were assessed with a Vario EL (Elementar Analysensysteme GmbH, Langenselbold, Germany) with an analytical precision of 0.01 g kg$^{-1}$ for C and N, respectively, according to DIN/ISO 13878:1998 and 10694:1996 [24,25].

Isotopic signature $\delta^{15}N$ was measured using EA-IRMS, i.e., an element analyzer coupled to an isotope ratio mass spectrometer (Flash-EA and DELTA V Advantage, Thermo Fisher Scientific GmbH, Dreieich, Germany). Measurements were calibrated through regression in three dimensions (signature × ion amount × time) against certified reference materials, which were treated and measured in the same way as the samples, and which covered the value ranges of the samples, i.e., $\delta^{15}N$: 1.18‰ to 19.6‰ air, sample weight: 0.3 mg to 1.0 mg (acetanilide #1, acetanilide #2, L-phenylalanine, Arndt Schimmelmann, Bloomington, IN, USA). Standard deviation of measured values against the certified value of the reference materials was less than 0.2‰ (n = 11) for $\delta^{15}N$. In accordance with Rocha et al. [8], the $^{15}N$ enrichment (atom % $^{15}N$ excess) in plant parts and soil were obtained by deducting the natural abundance. The latter was assessed in the not fertilized control treatments. The share of nitrogen derived from fertilizer (NdfF (%)) was calculated according to Equation (1) (in accordance with IAEA [26]).

$$\text{NdfF (\%)} = \text{atom \% }^{15}\text{N excess in sample}/\text{atom \% }^{15}\text{N excess in fertilizer} \times 100 \quad (1)$$

This results in the amount of fertilizer N in the corresponding plant fraction or in the soil according to:

$$\text{NdfF [g]} = \text{Ntot} \times \text{NdfF (\%)} \quad (2)$$

where Ntot describes the total amount of N (g) in the plant parts or the soil.

The amount of N derived from soil (NdfS) in the plant parts results from the subtraction of NdfF from Ntot:

$$\text{NdfS} = \text{Ntot - NdfF} \quad (3)$$

Calculation of FNU was then carried out according to IAEA [17]:

$$\text{FNU [\%]} = \text{NdfF in plant biomass}/\text{Nfert} \times 100 \quad (4)$$

where Nfert is the applied fertilizer N amount (g) and NdfF is the total N uptake (g) by plant biomass.

The fertilizer recovery rate in plant and soil ($^{15}NRR$) is calculated according to

$$^{15}\text{NRR [\%]} = 100/\text{Nfert} \times (\text{NdfFplant + NdfFsoil}) \quad (5)$$

where NdfFplant and NdfFsoil is nitrogen derived from fertilizer in plant biomass and soil, respectively.

*2.3. Statistical Analysis*

For the independent variables (weights, N and $^{15}$N amounts of plant fractions), mean values were tested for significant differences using two-way ANOVA (crop * fertilization), followed by Tukey HSD. For data that did not meet the requirements for ANOVA, the Kruskal–Wallis test followed by the Wilcoxon rank sum exact test was carried out. Dependent variables (changes in soil total N and soil $^{15}$N) were analyzed using *t*-test in the case of existing normal distribution. If the variables were not normally distributed, Kruskal–Wallis test (Dunn's post-hoc test with Bonferroni adjustment) was applied. Analysis was carried out with Rstudio statistical software, version 1.3.1056-1 [27].

**3. Results**

*3.1. Biomass and Soil*

At the start of the experiment, there were some inhibitions in plant germination, followed be a delayed plant development. This resulted in a slightly longer growing season compared to the field (154 days). N fertilization resulted in higher dry matter production of ABM and TBM of sorghum sudangrass (Table 2). For maize, a positive fertilization effect was only observed with regards to TBM yield. In contrast, fertilization did not affect growth of S + R of both crops. The ratio of ABM: S + R of sorghum sudangrass increased with N fertilization from 6.7 to 8.2. An increase in the ratio ABM:S + R of maize was not observed, and the ratio accounted for 7.3 with both fertilization treatments. Sorghum sudangrass achieved higher overall TBM yields compared to maize in both fertilization treatments, but ABM of sorghum sudangrass was higher than maize only in fertilization treatment N1. Extrapolated to one ha, the TBM yields obtained correspond to about 39 and 28 Mg*ha$^{-1}$ each for fertilized sorghum sudangrass and maize, respectively.

**Table 2.** Dry matter yields (g DM*bag$^{-1}$) and N content (%) of aboveground biomass (ABM), stubble + rootstock (S + R) and total biomass (TBM) for sorghum sudangrass and maize with fertilization treatment N0 (0 g N*bag$^{-1}$), and N1 (1.76 g N*bag$^{-1}$). Letters denote significant differences for each plant part between crops and fertilization treatments.

| Crops | | Sorghum Sudangrass | | | | Maize | | | |
|---|---|---|---|---|---|---|---|---|---|
| Fertilization Treatment | | N0 | | N1 | | N0 | | N1 | |
| ABM | (g DM*bag$^{-1}$) | 328.6 | a | 431.6 | b | 305.3 | c | 373.9 | ac |
| | N (%) | 0.61 | a | 0.64 | a | 0.58 | a | 0.58 | a |
| S + R | (g DM*bag$^{-1}$) | 55.7 | abc | 59.8 | abc | 48.4 | b | 64.9 | c |
| | N (%) | 0.24 | a | 0.31 | b | 0.19 | a | 0.23 | b |
| TBM | (g DM*bag$^{-1}$) | 384.3 | a | 491.4 | b | 353.7 | a | 438.8 | b |
| | N (%) | 0.55 | a | 0.59 | b | 0.53 | a | 0.53 | b |

For both sorghum sudangrass and maize, N fertilization resulted in higher N content in all plant fractions (Table 2). These were on average higher in N1 compared to N0. Differences between crops were not observed. The N content (%) of ABM on average was about two to three times higher than of S + R. Sorghum sudangrass absorbed more NdfF than maize in all plant fractions (Figure 1). Furthermore, no differences were observed between the crops in ABM, nor was there any fertilizer effect in ABM of the respective crops. In the S + R fraction, N fertilization led to increased Ntot compared to the unfertilized treatment N0. Without N fertilization, sorghum sudangrass accumulated more Ntot in S + R than maize in that fraction, while there were no differences in Ntot uptake of TBM of unfertilized plants. The only difference was observed for fertilized sorghum sudangrass, that took up more Ntot than unfertilized maize (Figure 1). Remaining amounts of NdfF in soil after harvest ranged between 12% after maize and 15% after sorghum sudangrass of the initially applied amount (1.76 g N*bag$^{-1}$) and was not different for the two crops.

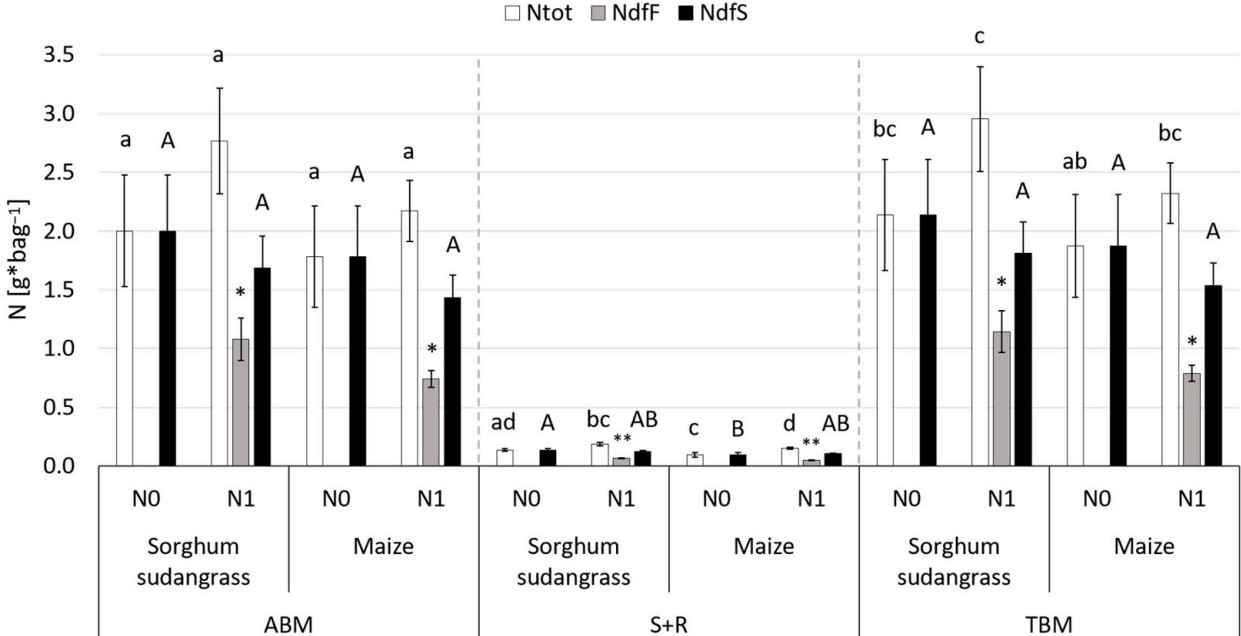

**Figure 1.** Total N uptake (Ntot), fertilizer N (NdfF) and soil N (NdfS) for plant parts of sorghum sudangrass and maize with the respective fertilization treatment (N0 and N1). Error bars show standard deviation. Lower case letters denote differences of Ntot between sorghum sudangrass and maize as well as fertilization treatments (N0 and N1) within respective plant parts. Upper case letters denote differences of NdfS between sorghum sudangrass and maize as well as fertilization treatments (N0 and N1) within respective plant parts. * and ** denote significant differences (at alpha = 0.5 and 0.1, respectively) of NdfF between sorghum sudangrass and maize within respective plant fractions.

### 3.2. Nitrogen Uptake Characteristics

The recovery rate ($^{15}$NRR) of Nfert (Table 3) was higher with sorghum sudangrass (81.1%) compared to maize (56.9%). TBM of sorghum sudangrass had a higher NdfF compared to maize (38.7% and 34.0%, respectively). Both plants accumulated more N than was applied through fertilization. The FNU is higher with sorghum sudangrass (65.0%) compared to maize (44.8%).

**Table 3.** Recovery rate ($^{15}$NRR) of fertilizer, not recovered fertilizer N amount (Nfert), share of fertilizer N on total N uptake by plants (NdfF), and plant utilization of fertilizer (FNU). Mean values displayed for fertilization treatment N1 (1.76 g N*bag$^{-1}$) of sorghum sudangrass and maize. TBM = total plant biomass. Letters denote significant differences between sorghum sudangrass and maize, respectively.

| Crops | Sorghum Sudangrass | | Maize | |
|---|---|---|---|---|
| Fertilizer recovery rate ($^{15}$NRR (%)) | 81.1 | a | 56.9 | b |
| Nfert not recovered (%) | 19.9 | a | 43.1 | b |
| Nitrogen in TBM derived from fertilizer (NdfF (%)) | 38.7 | a | 34.0 | b |
| Fertilizer nitrogen utilization of TBM (FNU (%)) | 65.0 | a | 44.8 | b |

## 4. Discussion

### 4.1. Observed Biomass Production

In the experiment, sorghum sudangrass produced yields of ABM corresponding to 34.2 Mg*ha$^{-1}$. This yield is somewhat high, yet still plausible compared to yield ranges in practice of 18.0 to 33.1 Mg*ha$^{-1}$ [28]. For maize, yields have been reported in the range of 15.8 to 22.7 Mg*ha$^{-1}$ [29]. Against high seed density of maize in the bag experiment, the rather high production of ABM (corresponding to 29.6 Mg*ha$^{-1}$) is plausible as well. The aim of the discussion is—in addition to the interpretation of differences between sorghum sudangrass and maize—also the examination of a possible transferability of the results of the bag experiment to the field level.

The S + R fraction was chosen in our study as the bulk of the crop and root residues and is not reported in this form by other studies. The fraction contains both a portion of the aboveground biomass (stubble) and a significant portion of the root biomass, the rootstock. The total root biomass for maize averages 31 g*plant$^{-1}$ [30]; thus, for our experiment, it should account for approximately 155 g*bag$^{-1}$ (with five plants per bag). Since the measured masses of the S + R fraction were min. 48 and max. 69 g*bag$^{-1}$, it must be assumed that a considerable part of the roots was not recorded. On the other hand, it is extremely difficult to draw conclusions about root development, especially root mass growth of a plant under field conditions, from observations made in pot experiments or bag experiments. This is due to the severely compromised soil conditions in the pots or bags. After all, roots could have been underdeveloped compared to the field, since fertilization was more direct and there was no need (and no space) to expand. For this reason, it is not so much the root masses as the N uptakes of fraction S + R that are discussed here. Other studies report N contents in the organic biomass of maize ranging from 0.65% to 0.98% [31,32]. On this basis, the observed N contents of ABM from maize in our experiment are plausible (0.58%). In the Lynch et al. study [32], the biomass is plant residues, and no further definition is provided as to which residues are involved. The authors also reported mean N contents of 1.09% in sorghum sudangrass residues. These contents are again a little higher than observed in our experiment for sorghum sudangrass (0.61% to 0.64%). The sometimes two- to threefold lower N contents in the fraction S + R of both sorghum sudangrass and maize might indicate N translocation from vegetative plant parts (e.g., stubble mass and rootstock) during grain filling.

### 4.2. NdfF and Implications for FNU

As already described in the introduction, studies on TBM of maize report NdfF ranging from 11% to 74% with a median of 31% [3–8]. In our experiment, NdfF of sorghum sudangrass and maize lies within this reported range (with 39% and 34%, respectively) and is comparable to the calculated median. Although NdfF was significantly higher for sorghum sudangrass than for maize, the magnitude was rather low.

FNU of maize is reported to range from 24% to 62% with a median of 43% (see Appendix A Table A1), which is close to observed FNU of cereal that ranges from 27% to 66% with a median of 46% (see Appendix A Table A1). In our study, FNU of maize (45%) is in line with that range and the median is similar.

Sorghum sudangrass, on the other hand, had a much higher FNU of 65%. In search of an explanation, differences in root growth are an obvious possibility, since roots play a key role in the specific use efficiency [33]. Clark [34] highlights the special growth performance and the good root system of sorghum sudangrass, as well as its adaptability to stressful conditions. However, in our experiment, there were no differences found in total root mass of sorghum sudangrass and maize. A difference in root architecture and/or speed of growth is highly possible. A higher growing rate of roots might lead to a higher utilization of fertilizer N before it is transported to deeper soil layers and/or possibly is lost. Therefore, Olson et al. and Olson and Swallow [11,12] underlined the importance of the time of fertilization for NdfF in crops. The authors stated that FNU will be higher if the fertilization takes place contemporary with the plant's demand for N. The authors, in

addition, speculated about a "nitrogen sink" in the soil that has to be filled up before plants can utilize the fertilizer N [12]. This assumption comes from the observation of higher FNU with higher N fertilization. Therefore, higher FNU with higher fertilization rates might be expected for sorghum sudangrass as well. When extrapolating those findings to the field level, one has to keep in mind the higher seed density of maize in our bag experiment. At field level and with typical seed density, the comparison of root growth might have resulted in different observations and should not be overestimated.

After harvest of sorghum sudangrass and maize, about 13.5% of the applied fertilizer N was recovered in the soil. This share is on the lower end of shares reported in other studies on maize, which range from 14% to 46% (median = 25). However, it can be assumed that fertilizer N that was not found in the soil material might have been transformed by microorganisms with an unknown magnitude (e.g., $NO_3$ to $N_2$) and subsequently lost to volatilization. At the end of our experiment, about 20% and 43% of fertilizer N was not accounted for with sorghum sudangrass and maize treatments, respectively. This is in line with findings of comparable studies that include the assessment of fertilizer N in TBM as well as in soil or that at least assessed the effect of possible sources of N loss (see Appendix A Table A1). Those studies report a fertilizer N loss of 29% (median) in experiments with maize, ranging from 15% to 76%. There are different reasons and explanations for a loss of fertilizer N in such experiments (see Appendix A Table A1). However, the quantification of those losses is reported to be specifically difficult [35,36]. Most studies assign the observed losses to, e.g., denitrification, leaching, or $NH_3$ volatilization, even though this was not quantified in the study. Our data show a strong relationship between fertilizer N loss and FNU with sorghum sudangrass and maize ($R^2 = 0.87$), which could be expected. However, the correlation does not provide full information on the causal relationship. From the data, it cannot be concluded whether higher fertilizer N loss reduced FNU of the plants or whether an inhibited FNU consequently resulted in higher fertilizer N losses. The magnitude of losses observed in our experiment are reasonable with regard to possible pathways. After irrigation, some leachate could be observed under the bags, which were placed directly on the ground of the greenhouse. However, irrigation was carried out very carefully to avoid leachate. It is still likely that NdfF was lost within leachate, especially after the early fertilizer application (no plant uptake in the initial period). Next to that, nitrification and denitrification are reported to be an important source for $N_2O$ losses from soils [33]. In contrast, other studies have stated that nitrification is probably less important for N losses of arable soils [37]. Kastori [38] observed up to 80 kg $N*ha^{-1}$ lost through volatilization during one growing and that $NH_3$ released from aboveground biomass can hold 52% to 73% of observed fertilizer N losses as well as 15% to 20% of FNU [3]. In our study, the initial procedure of mixing the soil substrate with the liquid fertilizer was rather intense compared to arable soils. Thus, we assume the initial microbial activity in the substrate increased significantly, resulting in higher $^{15}N$ losses due to nitrification and volatilization.

It must also be taken into account that the present study did not record and analyze the entire root mass. Root stock (including coarse roots) and stubble mass represent a great part of the crop residue that remains in the soil when whole crops are harvested for energy use. Because the assessment of fine roots is quite expensive and not always viewed as totally objective [39], we decided to limit our observations to the combined S + R fraction. However, it can be expected that the $^{15}N$ recovery and the observed fertilizer use efficiency would be higher if the entire root mass had been included.

In addition to in situ losses of nitrogen, sampling procedures and further treatment of samples contribute to nutrient loss. Possible errors can occur due to, e.g., dry matter loss during washing of roots, root respiration after sampling, C and N losses during drying, and storage of samples [40–43].

*4.3. Data Suitability for SOM Balance at Field Level*

One of our experimental questions was to what extent the data collected in our experiment are suitable to be used in SOM balance models. At this point, the disadvantages and advantages of a bag experiment in a greenhouse have to be discussed against the background of this specific question.

A clear shortcoming of bag experiments lies in the transferability of data to the field level. Differences in soil and climatic conditions can influence overall biomass production and root development as well as nutrient uptake of the plants. Therefore, the interpretation of results must be made with care. The different space availability, but also the different water and temperature regimes in bags most likely alter root development of a plant in comparison to the field. Next to effects on plant growth, this can lead to a misinterpretation of the rhizodeposition when transferring from the bag experiment to field level. At the same time, complete root coverage carries some risks for error [37]. Due to the high costs of complete root collection, its necessity should be discussed before the experiment. In addition, temperature regimes in a greenhouse and controlled water supply to the bags distinguishes such experiments from normal field conditions. Against this background, it is better to study plants under actual field conditions if the subject matter addresses the assessment of absolute values (e.g., C sequestration). However, bag experiments offer a great advantage if the subject matter addresses the assessment of relative values, such as, in our case, NdfF and FNU of one plant compared to another. Reasons for can be found in homogeneous and overall known experimental parameters (including soil). This leads to better comparability of results.

As already introduced, maize is a well-studied crop and parameters needed for SOM balancing are available to a great extent. In contrast to maize, considering data on sorghum sudangrass, especially the hybrid *sorghum bicolor* × *Sorghum sudanense*, the data situation looks much thinner. Since maize shows some physiological similarities to sorghum species, parameters of maize may be used as substitutes for sorghum in SOM balance models. Due to reported differences between maize and sorghum [14–17], we question the admissibility of this procedure. In our experiment, maize and sorghum sudangrass significantly differed with regard to NdfF and FNU (parameters needed for SOM balance). At the same time, observed data for maize are plausible and comparable with other studies. For this reason, we assume that the data collected for sorghum sudangrass on NdfF and FNU are equally plausible. However, we have to stress the fact that our experiment was carried out on only one soil substrate, with one fertilizer type and only two fertilizer levels. In addition, we did not record the amounts of N loss and N in fine roots. These are methodologically as difficult to capture in field experiments as in bag experiments. In future studies, N leachate, volatile N losses, and fine roots may receive a higher focus to better estimate the N pathways of one plant versus another. In addition, further experiments with different soil types, fertilizer types, fertilizer rates, different varieties and seed densities would further complete the dataset on sorghum sudangrass.

Nevertheless, for the first evaluation of sorghum sudangrass with SOM balance models, we recommend using the data presented here on NdfF and FNU for sorghum sudangrass and not relying on data from maize in this case. We base this recommendation on the significant differences compared to maize observed in our experiment and the physiological differences of the two plants mentioned in the literature. Those estimates, however, should be verified in long-term studies on sorghum sudangrass at field level. Against this background, long-term experiments have to be designed that deliver suitable data.

## 5. Conclusions

From the findings of our study, we conclude that sorghum sudangrass differs significantly from maize with regards to nitrogen utilization and FNU. In conclusion, parameters of maize should not be used as a substitute value in the SOM balance of sorghum sudangrass at field level. Since the observations made for maize are comparable to similar studies, we assume that the results for sorghum sudangrass are plausible as well. We therefore prefer to use the collected results of our experiment on sorghum sudangrass for the preliminary parametrization of this crop in SOM balance. However, a greenhouse experiment differs in conditions like soil and water conditions, climate, etc., compared to the field. With regards to the increasing importance of bioenergy plants, we hence highly suggest further studying crops like sorghum sudangrass, especially with regard to nitrogen utilization, N gas emission, and N leachate, both in the greenhouse and in long-term field studies. Our study is a first valuable contribution to this collection of data.

**Author Contributions:** Conceptualization, L.K. and C.B.; methodology, L.K.; software, L.K.; validation, L.K., C.B. and A.G.; formal analysis, L.K.; investigation, L.K.; resources, C.B.; data curation, L.K.; writing—original draft preparation, L.K.; writing—review and editing, C.B., A.G. and W.N.; visualization, L.K.; supervision, C.B. and A.G.; project administration, C.B.; funding acquisition, C.B. All authors have read and agreed to the published version of the manuscript.

**Funding:** This research was funded by FACHAGENTUR NACHWACHSENDE ROHSTOFFE e.V. (FNR), grant number 22402112 and the APC was funded by Justus Liebig University Giessen.

**Institutional Review Board Statement:** Not applicable.

**Informed Consent Statement:** Not applicable.

**Data Availability Statement:** Data from this study will be shared with justified request to the corresponding author.

**Acknowledgments:** We would like to thank the FACHAGENTUR NACHWACHSENDE ROHSTOFFE for funding the project as well as our project partners at the Humboldt University of Berlin (Christof Engels). Special thanks goes to Klemens Eckschmitt from JLU Gießen for coordinating the $^{15}$N analysis. We are also grateful to our staff in the greenhouse and laboratory.

**Conflicts of Interest:** The authors declare no conflict of interest. The funders had no role in the design of the study; in the collection, analyses, or interpretation of data; in the writing of the manuscript; or in the decision to publish the results.

## Appendix A

**Table A1.** Overview on key data from studies used for discussion (crops, parameters and specifications, fertilization, NdfF (%), FNU (%), and $^{15}$N, either remaining in soil or not recovered; Exp. = experiment).

| Crop | Parameters and Specifications | Fertilization | NdfF | FNU | $^{15}$N Remaining in Soil | $^{15}$N Lost or Not Recovered | References |
|---|---|---|---|---|---|---|---|
| Maize (irrigated in two experiments) | Leaves, stalks, and grain in both experiments; 240 cm soil depth in Exp. 1 | Exp. 1: 50, 100, and 150 kg N*ha$^{-1}$ as NH$_4$NO$_3$ at V3 stage; Exp. 2: 75 to 300 kg N*ha$^{-1}$ as (NH$_4$)$_2$SO$_4$ at V3 stage. | Exp. 1: 12 to 31% (at maturity); Exp. 2: 21 to 55% (at maturity) | Exp. 1: 48 to 53%; Exp. 2: 24 to 36% | Exp. 1: 24% (with 50 kg N rate) and 20% (with 100 and 150 kg N rate) | Exp. 1: 29% (mean) of which 52 to 73% lost by NH$_3$ volatilization of plants between blister and maturity (15 to 20% of FNU) [1]; Exp. 2: 64 to 76% | Francis, Schepers and Vigil (1993) [3] |
| Maize | Grain, stover, roots, and soil (after one growing season) | 124 kg N*ha$^{-1}$ before sowing (granular, incorporated) as (NH$_4$)$_2$SO$_4$ | Not assessed | 40% | 23% | 38% (due to leaching and denitrification) [2] | Harris et al. (1994) [4] |
| Maize (irrigated, 2 yr observation) | Grains, cobs, and stover | 50 kg N*ha$^{-1}$ and 150 kg N*ha$^{-1}$ before seeding as (NH$_4$)$_2$SO$_4$ | 11% (with 50 kg N rate); 30% (with 150 kg N rate) | 42% and 49% at 50 kg N rate and 150 kg N | After 1 year 30% with 50 kg N rate and 27% with 150 kg N rate | 17 to 18% (due to leaching and denitrification) [2] | Olson (1980) [5] |
| Maize | Aboveground biomass (at dent), (3 yr observation period and 3 different water regimes) | 0 kg, 125 kg, 251 kg, and 376 kg N*ha$^{-1}$ (surface-applied and raked in) as (NH$_4$)$_2$SO$_4$ | 43% to 74% | 43% to 62% | Not assessed | Only presented in diagrams, but rather high (due to detectability of soil $^{15}$N amounts) [2] | Porter (1995) [6] |
| Maize | Grains, stover, and litter (after first growing season) | 140 kg N*ha$^{-1}$ (30 kg at planting and 110 kg top-dressed at V5) as (NH$_4$)$_2$SO$_4$ (granular) | 21% in grain, 65% in stover, 33% in shoots | 35% with shoots and 4% with litter | 46% | 15% (2% due to NH$_3$ volatilization [1], rest rather due to leaching than N$_2$O) [2] | Rocha et al. (2019) [7] |

Table A1. *Cont.*

| Crop | Parameters and Specifications | Fertilization | NdfF | FNU | $^{15}$N Remaining in Soil | $^{15}$N Lost or Not Recovered | References |
|---|---|---|---|---|---|---|---|
| Maize | Grain, stover (1 plant per microplot), and soil (15 cm soil depth) | 200 kg N*ha$^{-1}$ as $(NH_4)_2SO_4$, 133 kg N*ha$^{-1}$, 10 days prior to planting (incorporated), second application at V6 (raked in) | Not assessed | 32% (with application before seeding) and 48% (with application at V6) | 15% (with application before seeding) and 14% (with application at V6) | 53% (with application before seeding) and 38% (with application at V6) (due to leaching) [2] | Seo, Meisinger, and Lee (2006) [8] |
| Wheat | grain and straw at ripe stage; soil (90 cm soil depth) | 50 kg N*ha$^{-1}$ before sowing as $(NH_4)_2SO_4$ and $KNO_3$ | 25% in average, (in wheat tops) | 47% in average, (in wheat tops) | 29% on average, mostly in organic form (>8%) | 19% in average | Ladd and Amato (1986) [9] |
| Winter wheat | Plant tops, large roots, soil (180 cm soil depth); 1 to 5 yr obs. period | 50 kg and 100 kg N*ha$^{-1}$ as $(NH_4)_2SO_4$ (incorporated in fall, surface-applied in spring) | 22% and 26% (with 50 kg N rate in fall or spring); 38% and 40% (with 100 kg N rate in fall or spring) | After 1 year 44% and 46% (with 50 kg N rate in fall or spring); 48% and 57% (with 100 kg N rate in fall or spring) | After 1 year 36% and 34% (with 50 kg N rate in fall or spring); 29% and 23% (with 100 kg N rate in fall or spring) | After 1 year 20% loss on average (due to leaching, but mainly denitrification) [2] | Olson et al. (1979) [10]; Olson and Swallow (1984) [11] |
| Barley | Grain, straw, stubble, and soil (70 cm soil depth) | About 140 kg N*ha$^{-1}$ as $NH_4NO_3$ (in a solution), 6 weeks after sowing | 36 to 48% | 34 to 47% Including weed | 34 to 37% | 18.1 to 27.6% | Glendining et al. (1997) [12] |
| Spring barley | Straw and grain at two sites (subsequent years) | 30 to 150 kg N*ha$^{-1}$ as $NH_4NO_3$ with $^{15}$N enrichment either at sowing or booting stage | 6% in average (being increased at booting stage for one site) | 27 to 41% (with application at sowing), 45 to 66% (with application at booting) | Not assessed | 59 to 73% (calculated by difference) | Tran and Tremblay (2000) [13] |

NdfF (%) = fertilizer N share on total N in plant biomass; FNU (%) = percentage of fertilizer N utilization by plant. Percentages are either reported in the reference study or have been calculated based on values shown (percentages are rounded up or down, respectively). [1] Measured/observed in reference study or previous study by same authors. [2] Assumed, based on literature discussion and experimental conditions.

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
