# Peer review of "Uptake of Fertilizer Nitrogen and Soil Nitrogen by Sorghum Sudangrass (Sorghum bicolor × Sorghum sudanense) in a Greenhouse Experiment with 15N-Labelled Ammonium Nitrate"

_soilsystems, doi:10.3390/soilsystems7030071_

Round 1

Reviewer 1 Report

General comments

This paper is very well written and the experiment has been designed very well.

I would have liked to see some results of SOC and SOM  in the two treatments. Additionally, shortcomings of bag experiments needs to be discussed better.

Specific comments

Abstract:

 Line 25, 26, 27, 28- please explain properly, expand SOM,

‘observations made with our control plant (maize), showed that the results are plausible…. the SOM balance at field level.’

‘Line 308 ‘-irrigation was carried out very careful to avoid leachate.’…carefully

Line 319-‘It must also be taken into account that the present study did not record and analyze  the entire root mass.’…Why is it so?

Author Response

Dear Colleague,

On behalf of all the co-authors, I would like to thank you for your effort in reviewing our article. Your comments were very valuable. We have discussed them in detail and implemented them where possible. We hope the implementation meets your expectations. Where we could not implement it, we have explained the reasons. In the attached document you will find the comments of all reviewers and our respective answers. In our list you are reviewer 1. In the uploaded, corrected article you can also see which changes were made. This does not include corrections to grammar and punctuation, which were also made.

We would be pleased if you are satisfied with the current state of the article and approve its publication.

With best regards.

Lucas Knebl

Reviewer 2 Report

well written and may be submitted after miner modification

Author Response

Dear Colleague,

On behalf of all the co-authors, I would like to thank you for your effort in reviewing our article. Your comments were very valuable. We have discussed them in detail and implemented them where possible. We hope the implementation meets your expectations. Where we could not implement it, we have explained the reasons. In the attached document you will find the comments of all reviewers and our respective answers. In our list you are reviewer 2. In the uploaded, corrected article you can also see which changes were made. This does not include corrections to grammar and punctuation, which were also made.

We would be pleased if you are satisfied with the current state of the article and approve its publication.

With best regards.

Lucas Knebl

Reviewer 3 Report

The topic of literary theory research is very meaningful, but there are still many problems in the article. It needs further revision by the author.

1. In the part of materials and methods, whether the one-year data of field experiment is representative.

2. In the part of materials and methods, there is a lack of relevant meteorological data.

3. In the part of discussion, there is a lack of comparative analysis with previous research results.

4. In the conclusion part, the conclusion of the article lacks data support.

5. In the part of references, the references of the article are too long, and the introduction and discussion written can not represent the latest research situation. It is suggested that the references of the last 3 years should be added.

Author Response

Dear Colleague,

On behalf of all the co-authors, I would like to thank you for your effort in reviewing our article. Your comments were very valuable. We have discussed them in detail and implemented them where possible. We hope the implementation meets your expectations. Where we could not implement it, we have explained the reasons. In the attached document you will find the comments of all reviewers and our respective answers. In our list you are reviewer 3. In the uploaded, corrected article you can also see which changes were made. This does not include corrections to grammar and punctuation, which were also made.

We would be pleased if you are satisfied with the current state of the article and approve its publication.

With best regards.

Lucas Knebl

Reviewer 4 Report

The paper is well-written and gives a clear summary of the study's findings. The study's findings are presented by the authors in a comprehensible manner, and they also address their implications.

There are a few potential restrictions on the results, though, that must be taken into account. First, because the study was done in a greenhouse, it's possible that the findings won't apply to real-world situations.

Second, because the trial was completed so quickly, the long-term impacts of fertilization and seed density on crop yield are unknown.

The temperature, humidity, and light intensity of the greenhouse, as well as other relevant factors, should be covered in more detail by the authors.

The authors should go into greater detail regarding the long-term impacts of seed density and fertilization on crop yield.

The findings imply that while a high seeding density can boost maize output, it can also cause plant competition for resources. This is a result of competition among plants for nutrients, water, and sunlight.

Correct the article's grammar and punctuation errors.

Line 27: correct therefore spelling

And many more.. 

The language is appropriate and I have no major objection, but yes there are some grammar mistakes.

Author Response

Dear Colleague,

On behalf of all the co-authors, I would like to thank you for your effort in reviewing our article. Your comments were very valuable. We have discussed them in detail and implemented them where possible. We hope the implementation meets your expectations. Where we could not implement it, we have explained the reasons. In the attached document you will find the comments of all reviewers and our respective answers. In our list you are reviewer 4. In the uploaded, corrected article you can also see which changes were made. This does not include corrections to grammar and punctuation, which were also made.

We would be pleased if you are satisfied with the current state of the article and approve its publication.

With best regards.

Lucas Knebl

Reviewer 5 Report

The manuscript subject corresponds to the journal. In general, the manuscript quite well reflects the current state of affairs in this field of science and presents new data. It is likely that dividing the Discussion into separate subchapters may be helpful for better presentation of the results.

However, I can recommend this article for publication in its current form.

Author Response

Dear Colleague,

On behalf of all the co-authors, I would like to thank you for your effort in reviewing our article. Your comments were very valuable. We hope the implementation meets your expectations. Where we could not implement it, we have explained the reasons. In the attached document you will find the comments of all reviewers and our respective answers. In our list you are reviewer 5. In the uploaded, corrected article you can also see which changes were made. This does not include corrections to grammar and punctuation, which were also made.

We would be pleased if you are satisfied with the current state of the article and approve its publication.

With best regards.

Lucas Knebl

Round 2

Reviewer 4 Report

The comments have been addressed